# Association of Family History and Life Habits in the Development of Colorectal Cancer: A Matched Case-Control Study in Mexico

**DOI:** 10.3390/ijerph18168633

**Published:** 2021-08-16

**Authors:** María del Pilar Ramírez-Díaz, María T. Alvarez-Bañuelos, Martha S. Robaina-Castellanos, Pedro P. Castro-Enríquez, Raúl E. Guzmán-García

**Affiliations:** 1Facultad de Nutrición, Universidad del Istmo, Juchitán 70102, Oaxaca, Mexico; mariadelpilard@bizendaa.unistmo.edu.mx; 2Instituto de Salud Pública, Universidad Veracruzana, Xalapa 91190, Veracruz, Mexico; pecastro@uv.mx; 3Instituto Nacional de Oncología y Radiobiología, La Habana 10400, Cuba; msrcastellanos@gmail.com; 4Centro Estatal de Cancerología “Dr. Miguel Dorantes Mesa”, Xalapa 91130, Veracruz, Mexico; raulguzman60@gmail.com

**Keywords:** colorectal cancer, epidemiology, case-control, family history, dietary

## Abstract

Colorectal cancer (CRC) is one of the most frequently diagnosed cancers and, as such, is important for public health. The increased incidence of this neoplasm is attributed to non-modifiable controls such as family history and modifiable variable behavioral risk factors involved in lifestyle like diets in Mexico. The presence of these factors is unknown. Therefore, the aim of this study was to evaluate family history and lifestyle factors associated with developing colorectal cancer in a Mexican population. Descriptive statistics and multivariate logistic regression were used to estimate the adjusted odds ratios (OR), as well as the 95% confidence intervals (CI). In this paper, significant differences were demonstrated between cases and controls. A family history of cancer (FHC) increased the probability of CRC [OR = 3.19 (95% CI: 1.81–5.60)]. The area of urban residence was found to be a protective factor compared to the rural area. This was also the case for frequent consumption of fruits [OR = 0.49 (95% CI: 0.28–0.88)], the frequent consumption of beef [OR = 2.95 (95% CI: 1.05–8.26)], pork [OR = 3.26 (95% CI: 1.34–7.90)], and region-typical fried food [OR = 2.79 (95% CI (1.32–5.89)]. These results provide additional evidence supporting the association of some CRC risk factors with family history of cancer, low fruit consumption, high consumption of red meat, and fried foods typical of the region of México. It is important to establish intervention methods, as well as genetic counseling to relatives of patients with CRC.

## 1. Introduction

According to the International Agency for Research on Cancer (IARC) in 2020, colorectal cancer (CRC) in Mexico is the third most incidental cancer, with a number of new cases at 7.6% and a mortality rate of 63.2 per 100,000 inhabitants (occurring more frequently in men) [1]. The etiology of CRC is complex and multi-causal, and is associated with both modifiable and non-modifiable factors. Race, ethnicity, sex, age, and hereditary-family history are among the non-modifiable factors. On the other hand, modifiable behavioral risk factors refer to lifestyle factors including diet, physical inactivity, obesity, and particularly, alcohol consumption, and tobacco use [2]. Genetic predisposition is one of the most explored factors in this type of cancer. Pathogenic germline variants in genes related to high-risk cancer have an impact on 2–8% of all cases. However, for mutations in high and moderate-penetrance known genes, the percentage increases from 6% to 10% [3,4]. In people with inherited syndromes, permanent risks range from 50–80% in the absence of preventive measures [5]. In cases with no germline mutations (15–30%), the presence of relatives with at least one first-degree relative (parents, siblings, or children), known in this cancer as familial aggregation, has been identified [6]. However, nearly 60–65% of CRC cases are sporadic (that is, they occur in individuals with no hereditary family history of CRC) and are attributed mainly to modifiable factors [7]. Lifestyle factors such as obesity, physical inactivity, alcohol consumption, tobacco use, frequent fat, red meat and sugar intake, and low fiber intake influence the development, treatment, and survival of CRC [8]. Currently, Mexico has been going through an important demographic, epidemiological, and nutritional transition. In 2018, at the national level, 75.2% of adults aged 20 and older were overweight and obese (mainly due to a poor diet and lack of physical activity) which have led to an increase in chronic diseases including cancer [9]. Veracruz is one of the states in Mexico with the highest cancer mortality rate, and future CRC projections show a considerable increase in the years to come [10]. However, information on the factors associated with this type of cancer is limited. Thus, in order to fill this gap in the literature pertaining to Mexico, this study aimed to assess the impact of family history cancer (FHC) and lifestyle in the development of CRC in the population.

## 2. Materials and Methods

We conducted a 2:1 (two controls for each case) age-and-sex matched case-control study in patients recruited from two tertiary care hospitals “Dr. Miguel Dorantes” Centro Estatal de Cancerología (State Cancer Center, CECAN by its Spanish acronym) and “Dr. Rafael Lucio” Centro de Altas Especialidades (High Specialties Center, CAE by its Spanish acronym) both belong to the Secretary of Health and serve the population of the same district. The participants were recruited in the period from December 2016 to May 2017.

The sample was analyzed using the case-control formula developed by Cotterchio [11]. The OR was 3.46 with a 95% confidence interval (CI) of 2.84–4.22, with a variable family history of CRC, which resulted in 98 cases and 196 controls. The cases were selected by consecutive sampling and the controls were age (±2 years) and sex matched.

### 2.1. Cases

The recruited cases were defined as CECAN patients with a confirmed histopathological diagnosis of CRC aged > 20 years. Patients with cognitive impairment or recurrent CRC were excluded, and those who decided to leave the study or did not have at least 80% of the information collected were also excluded.

### 2.2. Controls

The recruited controls were from another tertiary care hospital (CAE) to avoid overestimating the FHC variable. Patients without a diagnosis of any type of cancer and inflammatory bowel disease were included. They were age (±2 years) and sex matched. The exclusion criteria for the controls were the same as for the cases.

### 2.3. Data Collection

Data on risk factors associated with CRC were recorded in a research instrument integrated and validated (content) by experts: Stepwise approach to chronic noncommunicable diseases risk factor Surveillance and the global physical activity questionnaire (GPAQ) [12]. Additionally, we used the food frequency questionnaire (FFQ), which is used for the National Health and Nutrition Survey (ENSANUT, by its Spanish acronym) [13]; 29 foods associated with CRC were included. Then, ailments were divided into seven groups: fruits, vegetables, cereals, dairy products, red meat and sausages, hypercaloric, and fried foods typical of the region. Frequent consumption (≥3 times a week) was compared with infrequent consumption (<3 times a week). Data were collected by two nutrition experts who were previously trained to standardize the data collection.

### 2.4. Variables and Measurements

The habitual residence was considered rural if the place of residence had a population less than 2500 inhabitants, while it was considered urban if the place had more than 2500 inhabitants, according to the classification of Instituto Nacional de Estadística y Geografía (National Institute of Statistics and Geography, INEGI by its Spanish acronym) [14]. Other covariates included were occupation (agricultural and related activities vs. other activities). Socioeconomic status (SES) was classified according to the questionnaire developed by Asociación Mexicana de Agencias de Inteligencia de Mercado y Opinión Pública (Mexican Association of Market Intelligence and Public Opinion Agencies, AMAI by its Spanish acronym).

The families classified with an FHC eligible for the study were those that at least had one family member diagnosed with cancer and CRC in the first, second and third generation, one individual being a first degree relative of the other two and at least one individual diagnosed under 50 years [7].

Being overweight or obesity were also considered, according to both the World Health Organization (WHO) classification and the perception scale adapted by Stunkard [15]. Alcohol consumption was categorized by a standard drink (approximately 14 g of alcohol/day). Smoking was defined as one cigarette or more a day and it was categorized as either current, previous, and never smoker.

For the AP classification, the levels of Metabolic Equivalents (METS) were considered. For moderate and transport-related activities, four METs were assigned, while eight METs were assigned for vigorous activities. GPAQ uses an algorithm to determine two categories: inactive (<600 MET min week) and active (>600 MET min week). Sedentary behavior was determined by the hours spent sitting or reclining: >4 h were the cut-off point [16].

### 2.5. Statistical Analysis

A descriptive analysis was carried out for the sociodemographic variables, which were represented by frequencies and percentages. The univariate analysis for the categorical variables was performed using the X2 test and Fisher’s exact test. Factors with a value of *p* < 0.05 in the univariate analysis were included as parameters for the bivariate and multivariate analysis by means of logistic regression, including the variables associated with CRC (family history, area of residence, sedentary lifestyle, and diet). The results of the multivariate analysis were expressed as OR with their respective 95% confidence intervals (CI). Values of *p* < 0.05 were considered statistically significant. The Hosmer and Lemeshow test > 0.05 was used for goodness-of-fit considering a 95% confidence interval were estimated. Analyses were carried out using the statistical software SPSS, version 25.0 (IBM Inc., New York, NY, USA).

### 2.6. Ethical Considerations

This study was approved by the Research Ethics Committee of CECAN, Ministry of Health of Veracruz, registration number C.E.I./2017/019. After obtaining informed consent, cases and controls were interviewed to ensure privacy, in a similar interview setting, with the same duration in both groups.

## 3. Results

A total of 98 cases and 196 controls were included in the study. The greatest age distribution was found in the range of 50 to 59 years, 55.8 ± 14.8 years. In terms of sex, it was more prevalent among men (59.2%) compared to women (40.8%) in both groups. Most of the cases and controls studied live in urban areas (62.2% and 74.5% respectively). More than 50% of the cases and controls were married and the main occupations were housewife and peasant farmer in both groups. Regarding the socioeconomic level, the highest proportion in both groups was found in the extreme poverty category (39.8% and 47.4%, respectively) (Table 1).

FHC of any type was higher in the cases (60.2%) compared to the controls (27%), showing statistically significant differences (*p* < 0.001). Family aggregation in the second degree was higher in the cases (52.5%), (*p* < 0.001). Consequently, for the family history of CRC, the cases presented a higher prevalence, with a high statistically significant difference compared to the controls (*p* < 0.001). Similarly, the family history of CRC in first and second degree was higher for the cases. revealing statistically significant differences (*p* = 0.012 and *p* = 0.044, respectively). It is important to mention that the total number of relatives with an FHC and CRC in first, second, and third degree was variable; there was family aggregation (Table 2).

When analyzing the lifestyle variables related to CRC of the cases and controls, sedentary lifestyle showed statistically significant differences (*p* = 0.032), as in the categories reusing cooking oil (*p* = 0.015), fruit, beef, pork, and fried foods typical of the region intake (*p* < 0.001) (Table 3).

Finally, Figure 1 shows the association of bivariate analysis and multivariate analysis adjusted by logistic regression for FHC and lifestyle factors associated with CRC. FHC of any type of cancer increases the probability of developing CRC [OR = 3.19 (95% CI: 1.81–5.60)]. The urban residence area turned out to be a protective factor compared to the rural residence area. Similarly, frequent fruit intake (≥3 times a week) was associated as a protective factor [OR = 0.49 (95% CI 0.28–0.88)] to develop CRC, whereas frequent intake of beef (≥3 times a week) [OR = 2.95 (95% CI: 1.05–8.26)], pork [OR = 3.26 (95% CI: 1.34–7.90)] and typical fried foods of the region [OR = 2.79 (95% CI (1.32–5.89)] were associated as risk factors that increase the probability of developing CRC. After data adjustment, there was not significant association for sedentary behavior and reusing cooking oil.

## 4. Discussion

A family history of CRC with a wide spectrum of coaggregation of cancers could significantly increase the risk, which could be considered as a clear sign of this neoplasm, as well as a series of behavioral risk factors associated with lifestyles, such as sedentary lifestyle and diet characteristics. It is our understanding that this is the first study of this type that has been conducted in a Mexican population, which denotes that the FHC and CRC, including a combination of family history in an affected first degree relative (or even another second degree relative) showed statistically significant differences between both groups. In this sense, FHC was associated as a risk factor for developing CRC [OR = 3.19 (95% CI: 1.81–5.60)]. These results coincide with those recently reported [OR 3.2, (95% CI: 1.4–7.6)] [17]. Previous studies recognize family aggregation of cancer and CRC as an important factor. A recent meta-analysis suggested the risk association between family history of CRC in first degree relatives with a relative risk of 1.87 (*p* < 0.001) [18].

Other variables considered in this research were the risk factors for CRC. In the first place, sedentary habits were significantly associated with CRC in the bivariate analysis. However, in the multivariate analysis the association was inconsistent. This result is added to other previous studies where a sedentary lifestyle is significantly associated as a risk factor [19]. However, a well-defined consensus on measuring or classifying sedentary lifestyle from an observational point of view has not been reached. A recent prospective study evaluated the time in hours (>7 h) spent in front of a television, which has been considered a good predictor, showing a statistically significant association with the CRC risk, even independently of family history, increasing the risk in subgroups with higher BMI, smoking, and lower physical activity [20]. This result could highlight the importance of their study since a sedentary lifestyle is considered an independent predictor of metabolic risk with its own implications.

The coherence of our findings regarding the characteristics of the diet strengthens the fact that they are considered as important factors that have an impact on CRC. Our research evidenced the higher consumption of fruits as a protective factor in the development of CRC, which is consistent with a series that recently demonstrated the importance of fruit and vegetable intake as protective factors for this type of cancer [21]. Similarly, a diet poor in fruits and the presence of high-risk adenoma polyps were significantly associated with increasing the probability to develop CRC [22]. Since both vegetables and fruits are considered good sources of fiber, folic acid, B vitamins, minerals, and antioxidants, their protective properties could lie in the decreased gastrointestinal transit time, which leads to reduced concentrations of carcinogenic compounds [23]. Even more, the vegetarian pattern has been associated as an important factor for preventing CRC [24].

Based on the results yielded by the multivariate analysis, the consumption of beef and pork were significantly associated as risk factors for CRC. This result is largely allied to the epidemiological studies reported by different countries [25]. Increasing evidence reinforces and strengthens that consumption of red and processed meat has been considered one of the dietary factors most associated with CRC due to the activation of receptors in the epithelium, causing an inflammatory process, as well as the compounds generated by being cooked at high temperatures as heterocyclic aromatic amines and N-nitroso compounds [23]. In Mexico, the higher consumption of meat is influenced by high economic income and, despite being a middle-income country, a considerable consumption of red meat (mainly in young adults) has been reported. Although consumption of meat is lower compared to other countries in America, such as the United States and Canada, the nutritional transition and acculturation could increase consumption and, thus, the risk of CRC [26].

Most of the existing studies suggest that an interaction of high fat intake can produce changes in the composition and function of the microbiota, causing changes with inflammatory effects implicated in intestinal tumorigenesis [23,27]. Based on our results, the consumption of typical foods in the region that go through a frying process was associated as a contributing factor to the development of CRC. In contrast, a study by Wang et al. correlated the consumption of fried foods with colon and rectal cancer, but it was only significant for colon cancer [28]. This result is not very consistent, since this specific food group has not been studied independently. However, studies on eating patterns associated with CRC have shown that the eating pattern, characterized mainly by the consumption of red meat, processed foods, and saturated fat, has been associated as a risk factor for CRC [29].

Although the incidence of CRC is more frequent in developed countries with high income than in developing countries: the morbidity and mortality rates of this disease increased in countries with fewer resources, such as Mexico. In this sense, our research significantly associated residential areas urban as a protective factor for CRC. Although this result was unexpected, it has already been previously identified. In addition, there is evidence that residents of rural communities are more likely to have a late diagnosis [30].

Mexico is currently going through an important epidemiological and nutritional transition represented by the joint prevalence of overweight and obesity (75.2%), high concentrations of cholesterol and high triglycerides (32.7%), diabetes (10.3%), and hypertension (18.4%) due to an inadequate diet characterized by low consumption of fiber, added sugars and saturated fats [9,26]. These factors have also been associated with CRC, which could explain the upward trend in the number of cases [25].

Studies on FHC and lifestyle factors associated with CRC are limited in Mexico. Regardless of the findings of this research, future studies are needed to explore and conduct in depth studies of the risk factors by improving the methodological limitations of the retrospective case-control studies, such as memory and information bias, to implement preventive measures adapted to the Mexican context with this type of cancer.

## 5. Conclusions

The study shows that family aggregation continues being a clear sign of CRC and highlights the importance of the shared role played by the place of residence, with some lifestyle habits, such as sedentary lifestyle, and characteristics of the diet, such as low consumption of fruits, high consumption of red meat (beef and pork), as well as fried foods typical of the region. It should be noted that typical Mexican dishes are cooked with high fat content, in addition to the reuse of oil, reflecting the importance of food culture in disease processes.

On the other hand, it would be important to further explore the interactions between family history and environmental factors, which could modulate the risk of developing CRC in any direction, in order to guide public health prevention strategies, as well as monitoring the identified families that are treated by both institutions through genetic counseling to prevent or, otherwise, make an early diagnosis.

## Figures and Tables

**Figure 1 ijerph-18-08633-f001:**
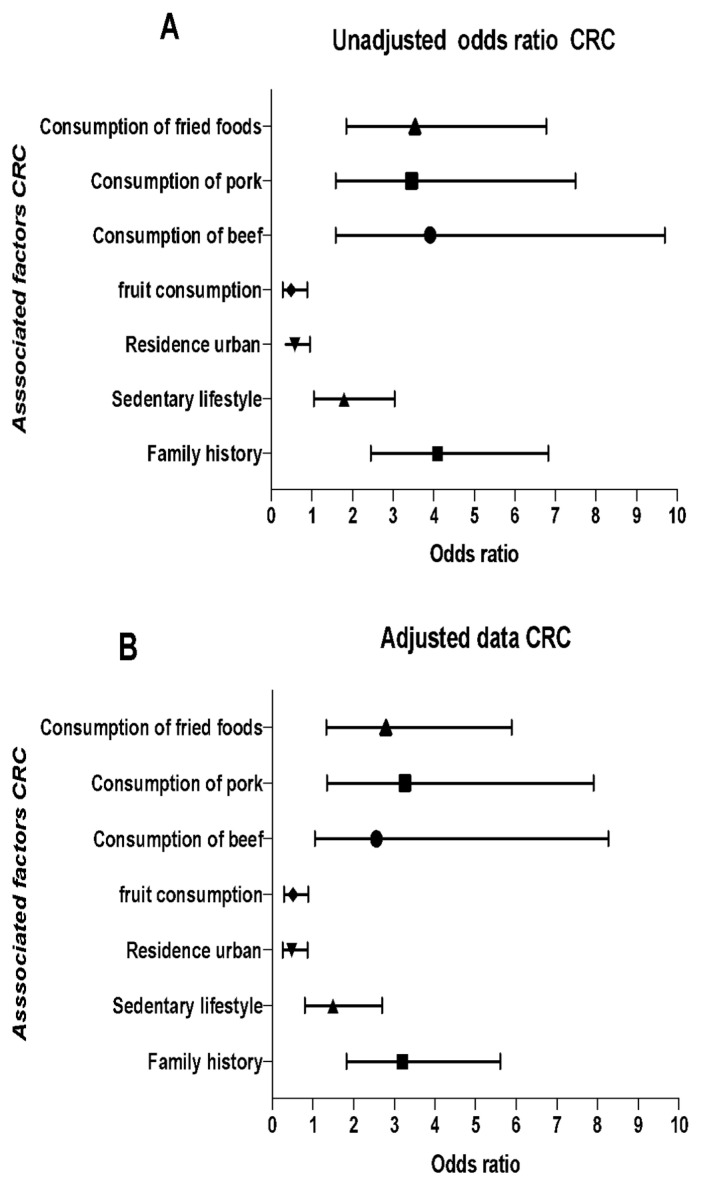
Multivariate analysis. History, family, and lifestyle factors associated with colorectal cancer. (**A**) Models unadjusted and (**B**) adjusted for the following variables: area of origin, marital status, sedentary lifestyle, consumption of fruits, beef, pork, and fried foods typical of the region by the “forward: conditional” method. In this method, the odds ratio is not shown for the variables that served in the adjustment. OR, ratio of odds (95%); CI, confidence interval (95%).

**Table 1 ijerph-18-08633-t001:** Sociodemographic characteristics of the cases and controls.

Variable	Cases		Controls	
	*n* = 98 (%)		*n* = 196 (%)	
Age (year)				
<40	11	(11.2)	23	(11.7)
40–49	23	(23.5)	45	(23)
50–59	24	(24.5)	47	(24)
60–69	22	(22.4)	45	(23)
≥70	18	(18.4)	36	(18.4)
Sex				
Man	58	(59.2)	116	(59.2)
Woman	40	(40.8)	80	(40.8)
Residence area				
Rural	37	(37.8)	50	(25.5)
Urban	61	(62.2)	146	(74.5)
Civil status				
Single	24	(24.5)	29	(14.8)
Married	50	(51)	99	(50.5)
Divorced	4	(4.1)	5	(2.6)
Widower	9	(9.2)	16	(8.2)
Free Union	11	(11.2)	47	(24)
Occupation ^a^				
Housewife	25	(25.5)	70	(35.7)
Peasant Farmer	21	(21.4)	54	(27.6)
Others	50	(51)	50	(25.5)
unemployed	2	(2)	22	(11.2)
Socioeconomic level				
High	3	(3.1)	6	(3.1)
Medium high	6	86.1)	9	(4.6)
Medium	3	(3.1)	8	(4.1)
Medium low	13	(13.3)	19	(9.7)
Low	17	(17.3)	36	(18.4)
Under extreme	39	(39.8)	93	(47.4)
Very extreme	17	(17.3)	25	(12.8)

^a^ Occupation prior to the diagnosis of the cases.

**Table 2 ijerph-18-08633-t002:** Family history of any type of cancer and colorectal cancer.

Variables	Cases		Controls		*p* Value
	*n* = 98 (%)		*n* = 196 (%)		
Family history of any cancer					
Yes	59	(60.2)	53	(27)	<0.001 *
No	39	(39.8)	143	(73)	
Cancer in type of relative					
1st Degree	24	(39.3)	30	(53.6)	0.055
2nd Degree	32	(52.5)	23	(41)	<0.001 *
3rd Degree Total	5 61	(8.2) (100)	3 56	(5.4) (100)	0.122 ^a^
Family history of CRC					
Yes	8	(8.2)	1	(0.5)	<0.001 ^a,^*
No	90	(91.8)	195	(99.5)	
CRC in type of relatives					
1st Degree	4	(40)	0	(0)	0.012 ^a,^*
2nd Degree	4	(40)	0	(100)	0.044 ^a,^*
3rd Degree Total	2 10	(20) (100)	1 1	(0) (100)	0.111 ^a^

^a^ Comparison of proportions by χ^2^ (Fisher’s exact test was used when necessary). * *p* value < 0.05.

**Table 3 ijerph-18-08633-t003:** Lifestyle factors in case and controls.

Variable	Category	Cases		Controls		*p* Value
		*n* = 98 (%)		*n* = 196 (%)		
Muscle mass index (BMI)						
Classification BMI	Low weight	5	(5.1)	26	(13.3)	0.170 ^a^
	Normal	51	(52)	107	(54.6)	
	Overweight	38	(38.8)	58	(26.6)	
	Obesity	4	(4.1)	5	(2.6)	
Smoke						
Exposure to cigarette smoke	No exposure	81	(82.7)	144	(73.5)	0.076 ^a^
	Active	0	(0)	21	(10.7)	
	Passive	17	(17.3)	31	(15.8)	
Ex-smokers and smokers ^b^	Yes	38	(38.8)	63	(32.1)	0.259
	No	60	(62.2)	133	(67.9)	
Cigars smoked daily ^c^						
	<4	12	(12.2)	19	(9.7)	0.531
	4 a 21	13	(13.3)	27	(13.8)	
	>21	13	(13.3)	17	(8.7)	
Alcohol						
Consumption of alcohol	Yes	66	(67.3)	114	(58.2)	0.120
	No	32	(32.7)	82	(41.8)	
Physical activity level						
	Inactive (<600 METs ^d^)	37	(37.8)	73	(37.2)	0.932
	Active (>600 METs ^d^)	61	(62.2)	123	(62.8)	
Sedentary	<4 h	64	(65.3)	151	(77)	0.032 *
	>4 h	34	(34.7)	45	(23)	
Feeding						
Reusing cooking oil	Yes	38	(38.8)	49	(25)	0.015 *
	No	60	(61.2)	147	(75)	
Vegetables Consumption	Frequently	44	(44.9)	108	(55.1)	0.092
	Infrequently	54	(55.1)	88	(44.9)	
Fruit consumption	Frequently	52	(53.1)	138	(70.4)	<0.001 *
	Infrequently	46	(46.9)	58	(29.6)	
Fish consumption	Frequently	8	(8.2)	7	(3.6)	0.092
	Infrequently	90	(91.8)	189	(96.4)	
Chicken consumption	Frequently	48	(49)	80	(40.8)	0.183
	Infrequently	50	(51)	116	(59.2)	
Beef consumption	Frequently	14	(14.3)	8	(4.1)	<0.001 *
	Infrequently	84	(85.7)	188	(95.9)	
Pork consumption	Frequently	18	(18.5)	12	(6.1)	<0.001 *
	Infrequently	80	(81.6)	184	(93.9)	
Sausage consumption	Frequently	4	(4.1)	11	(5.6)	0.574 ^a^
	Infrequently	94	(95.9)	185	(94.4)	
Ham consumption	Frequently	12	(12.2)	15	(7.7)	0.199
	Infrequently	86	(87.8)	181	(92.3)	
Consumption of fried foods typical of the region in Mexico	Frequently	27	(27.6)	20	(10.2)	<0.001 *
	Infrequently	71	(72.4)	176	(89.9)	

^a^ Comparison of proportions by χ^2^ (Fisher’s exact test was used when necessary). * *p* value < 0.05. ^b^ Ex-smokers aged 15 or younger who have quit smoking. ^c^ Number of cigarettes that ex-smokers smoked per day. ^d^ METs/wk: Metabolic equivalents minutes per week.

## Data Availability

The data used in this research are protected in the Public Health Institute of the Veracruzana University and may be requested from the corresponding author upon request.

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
