# Peer review of "Association of Family History and Life Habits in the Development of Colorectal Cancer: A Matched Case-Control Study in Mexico"

_ijerph, 2021, doi:10.3390/ijerph18168633_

Round 1
Reviewer 1 Report
Authors of the article investigated risk factors of colorectal cancer (CRC) within CRC patients from Mexico.
Major issues:
- The findings of the study are basically not novel, except for region-specific new data which should be at least mentioned within the conclusion.
- Why did authors recruited CRC patients and healthy controls at two different institutes? Were these individuals selected from the same district? This should be clarified in Methods.
- Abstract should be improved. There is no aim, Mexico was mentioned only at the end of it. The following sentence should be improved: "The increasingly incidence of this neoplasm is attributed to non-modifiable (family history) and modifiable (environmental factors involved in lifestyle such as diet) END IS MISSING."
Minor issues:
- Please, remove "Tilte" from the title.
- What software was used for data analysis? Please, include in methods.
- Please clarify sentence: "In terms of sex, it was more prevalent among men (59.2%) compared to women (40.8%)." CRC or control or both?
- Table 2: Please, use percenetages relative to the complate number of groups.
- Table 3: Please, fix the few Spanish words to English.
Author Response
Authors Comments
We are very grateful for the reviews provided by the editors and each of the external reviewers of this manuscript. The comments are encouraging and provided valuable information to refine your content and analysis. See below here and in the manuscript, in blue, our response to comments.
Revisor 1
Major issues:
- The findings of the study are basically not novel, except for region-specific new data which should be at least mentioned within the conclusion.
A: The specific results are novel, for the region of Mexico they were included in the conclusions.
- Why did authors recruited CRC patients and healthy controls at two different institutes? Were these individuals selected from the same district? This should be clarified in Methods.
A: The controls were selected from the same population as both hospitals belonged to the same district and Institutes of Health. This was clarified in Methods.
- Abstract should be improved. There is no aim, Mexico was mentioned only at the end of it. The following sentence should be improved: "The increasingly incidence of this neoplasm is attributed to non-modifiable (family history) and modifiable (environmental factors involved in lifestyle such as diet) END IS MISSING."
A: Corrections to the abstract were made, the objective was included, and Mexico is included in the abstract and in the conclusions.
Minor issues:
- Please, remove "Tilte" from the title.
A: the word title was deleted.
- What software was used for data analysis? Please, include in methods.
A: the software used was included in methods.
- Please clarify sentence: "In terms of sex, it was more prevalent among men (59.2%) compared to women (40.8%)." CRC or control or both?
A: sentence was corrected.
- Table 2: Please, use percentages relative to the complete number of groups.
A: the percentage of the total group was included.
- Table 3: Please, fix the few Spanish words to English.
A: words were corrected from Spanish to English.
Additional comment
The article was extensively reviewed for English and grammatical errors by a native-English speaking colleague. Authors made corrections to all suggestions made by the reviewers.

Reviewer 2 Report
In the manuscript entitled " Association of family history and life habits in the development of colorectal cancer: a matched case-control study in Mexico ", Ramírez-Díaz et al, showed the significant association of incidence of CRC risk factors (low fruit consumption, high meat and fried riched meals) with the family history among Méxician CRC diagnosed patients. They concluded that it is fundamental to establish prevention, as well as genetic counselling to relatives of CRC patients.
Overall, the study was conducted in an appropriate way, and the content is highly on-demand. however, the English and writing style are required extensive editing to be suitable for publication in the current journal.
Minor revisions:
The title:
The Word (Title) in the title sentence is a typo? Or the author meant something?
Abstract:
Line 12: please replace (increasingly) with the more appropriate one, such as (increased).
Line 18-20: the sentence is not completed, while the 18 frequent consumption of beef [OR = 2.95 (95% CI: 1.05-8.26)], pork [OR = 3.26 (95% CI: 1.34-7.90)] 19 and region-typical fried food [OR = 2.79 (95% CI (1.32-5.89)]??Then what?
Keywords:
It’s not suitable to mention (colon cancer and rectum cancer) types separately, as the study has been conducted on combined (Colorectal cancer). Accordingly, please remove these two keywords.
Introduction:
Lines 56 and 63: the portion (2: 1) in line 56 while in 63 is (1:2)! Is this right? If so what is the difference and what the author meant by using this portion?
Line 27-30: here the incidence rate were mentioned, however, it's highly recommended to show the percentage as it's easier for the reader.
Line 35: please remove the article (a) before (high-risk)
Line 54: The full stop is missed.
Materials and Methods:
Line 60: Please replace the verb (calculated) to (analysed).
Lines 69-73: Which kind of patients? Is it possible to consider other pathologic conditions as a control in such study?
Line 75: please replace (colorectal cancer) with (CRC).
Line 80-84: the word (food) is repeated more than three times, please change it with another synonym.
Results:
Overall, results are explained briefly, it's highly recommended to provide more details.
Line 129-130: the sentence is not correct rephrase it, please.
Line 139-141:again the repetition issue! (Family cancer history) please rewrite with another appropriate word.
Table 2, The titles in (Cases) and (Controls) the (n) or sample number is not written here while in Table 1 they were mentioned! Please unify providing information.
Author Response
Authors Comments
We are very grateful for the reviews provided by each of the external reviewers of this manuscript. The comments are encouraging and provided valuable information to refine your content and analysis. See below here and in the manuscript, in blue, our response to comments.
Revisor 2
Overall, the study was conducted in an appropriate way, and the content is highly on-demand. however, the English and writing style are required extensive editing to be suitable for publication in the current journal.
The article was extensively reviewed for English and grammatical errors by a native-English speaking colleague. Authors made corrections to all suggestions made by the reviewers.
Line 12: replace (more and more) with the most appropriate, such as (increased).
A: the word was changed.
Line 18-20: the sentence is not completed, while 18 frequent consumption of beef [OR = 2.95 (95% CI: 1.05-8.26)], pork [OR = 3.26 (CI 95%: 1.34-7.90)] 19 and typical fried food of the region [OR = 2.79 (95% CI (1.32-5.89)] ?? So what?
A: the prayer was completed.
Keywords:
It is not appropriate to mention the types (colon and rectal cancer) separately, since the study was done in combination (colorectal cancer). Consequently, remove these two keywords.
A: it was corrected, the two keywords were subtitled.
Introduction:
Lines 56 and 63: the portion (2: 1) in line 56 while in 63 is (1: 2)! Is this correct? If so, what is the difference and what did the author mean by using this part?
A: the error was run.
Line 27-30: the incidence rate was mentioned here, however, it is highly recommended to show the percentage as it is easier for the reader.
A: the suggestion is accepted, it changed by percentage
Line 35: remove item (a) before (high risk)
A: item (a) was removed.
A: suggestion is accepted.
Lines 69-73: What kind of patients? Is it possible to consider other pathological conditions as controls in this study?
A: the criteria were included all the inclusion criteria.
Line 75: replace (colorectal cancer) with (CRC).
A: it is changed to the acronym.
Line 80-84: the word (food) is repeated more than three times, change it to another synonym.
A: synonyms were used.
Results:
In general, the results are briefly explained, it is strongly recommended to provide more details.
A: results are expanded.
Line 129-130: the sentence is not correct, please modify it.
A: sentence was corrected.
Line 139-141: again the problem of repetition! (Family history of cancer) please rewrite with another appropriate word.
A: the words are changed.
Table 2, Titles in (Cases) and (Controls) the (n) or sample number are not written here, while in Table 1 they were mentioned. Unify by providing information.
A: the tables were unified.
In addition to the above comments, all spelling and grammatical errors pointed out by the reviewers have been corrected.
We look forward to hearing from you in due time regarding our submission and to respond to any further questions and comments you may have.
Sincerely,
Corresponding author

Round 2
Reviewer 1 Report
Authors addressed all my concerns.